# Environmental Hazards: A Coverage Response Approach

**Paul J. Croft**

School of Environmental & Sustainability Sciences, College of Natural, Applied, and Health Sciences, Kean University, Union, NJ 07083, USA; pcroft@kean.edu or pkwx87@gmail.com; Tel.: +1-908-737-4600

**Abstract:** The rapid rise and implementation of Smart Systems (i.e., multi-functional observation and platform systems that depict settings and/or identify situations or features of interest, often in real-time) has inversely paralleled and readily exposed the reduced capacity of human and societal systems to effectively respond to environmental hazards. This overarching review and essay explores the complex set of interactions found among Smart, Societal, and Environmental Systems. The resulting rise in the poorly performing response solutions to environmental hazards that has occurred despite best practices, detailed forecast information, and the use and application of real-time in situ observational platforms are considered. The application of Smart Systems, relevant architectures, and ever-increasing numbers of applications and tools development by individuals as they interact with Smart Systems offers a means to ameliorate and resolve confounding found among all of the interdependent Systems. The interactions of human systems with environmental hazards further expose society's complex operational vulnerabilities and gaps in response to such threats. An examination of decision-making, the auto-reactive nature of responses before, during, and after environmental hazards; and the lack of scalability and comparability are presented with regard to the prospects of applying probabilistic methods, cross-scale time and space domains; anticipated impacts, and the need to account for multimodal actions and reactions—including psycho-social contributions. Assimilation of these concepts and principles in Smart System architectures, applications, and tools is essential to ensure future viability and functionalities with regard to environmental hazards and to produce an effective set of societal engagement responses. Achieving the promise of Smart Systems relative to environmental hazards will require an extensive transdisciplinary approach to tie psycho-social behaviors directly with non-human components and systems in order to close actionable gaps in response. Pathways to achieve a more comprehensive understanding are given for consideration by the wide diversity of disciplines necessary to move forward in Smart Systems as tied with the societal response to environmental hazards.

**Keywords:** environmental hazards; smart systems; psycho-social; smart cities; emergency response; vulnerability; society; forecast; deterministic; observations

## 1. Introduction

The much anticipated value of (e.g., Pew Report, [1]) and rapid advancement in the capabilities [2,3], prevalence [4], and synthesis of Smart Systems within the cultural landscape [5–7] and within many Smart Cities on a global scale is well documented [8]. Smart Systems allow for decision-making using data and/or analyses from monitoring platforms or equipment for the prediction and verification of various quantities or outcomes according to intended purposes or user-defined functionalities. The promise and delivery of improved quality of life with social participation [9], and the role of Smart Systems in creating more efficient, effective [10], and more sustainability and leisure or health-oriented societies [11–13] is evident. The use and/or distribution

of resources (e.g., water), whether politically motivated or otherwise [14,15] as related to policy, are also laudable. Further, the combination of real-time in situ observational platforms [16] that include immediate dissemination offer the public access to view and monitor events or situations of interest continuously. Thus the utility and complexities associated with the application of Smart Systems to a wide variety of situations or intents (e.g., water quality) becomes apparent [17]. Whether intended for additional purposes or not, Smart Systems are applied "at-will" by commercial, industrial or manufacturing [18] interests as well as individual users and governments.

Concurrently the prevalence and increasing preponderance of significant environmental hazards worldwide have increased the burden they impose [19] on individuals and populations. This has brought into question the efficacy of Smart Systems in meeting multiple, increasing, and recurring losses, both economic and human. Together the increased burden of hazards and efficacy of Smart Systems raise the question that if interest exists in evolving more innovative business models [20], then why not with emergency response models linking interactive human responses with environmental hazards? What Smart Systems architectures [21] and applications [22], or components of the same, are missing? How are the uses, interpretations, and interactive responses of individuals [23], groups or crowds [24], and government agencies [25] being incorporated with such systems? These questions relate directly to applications designed for detection and/or diagnosis of flooding [26], assessment of urban climate monitoring [27], and earthquakes [28] with or without inclusion of online and Social Media information generated by the public.

In tandem are the observed shifts in the actual response of populations and governments to hazards (e.g., flooding, [29]] as tracked before, during, and after events. With the advent of user created content (e.g., as connected with volunteered geographic information [30]), and the associated high costs of rescue and recovery operations (e.g., emergency rescue, recovery, rebuilding, and insurance); repeated hazards pose a strain on the response systems and supporting infrastructure. These are due to user interpretations and expectations based upon created content that does not have a clear relationship to repeated hazards or responses. Thus additional questions arise as to the functionalities of Smart Systems in the advance education, preparation, and warning process. When facing increasing environmental hazards in large population areas [31], the combination of public responses to perceived harm [32]—regardless of the frequency, recurrence, or intensity of a particular hazard [33,34]—creates uncertainty when applied to locations with specific yet variable characteristics.

For these reasons multiple responses to environmental hazards [13,35] are employed by local governments before, during, and after events in and near urban and suburban regions often with large population densities amid widely divergent demographics. The modalities and effectiveness of these responses include sheltering in-place versus mandatory evacuation [36], steps to reduce or limit impacts rather than prevention (i.e., "expected loss" thresholds), and a halting of emergency response activation (i.e., when responders lives are endangered) during an event. Afterwards the "lock-down" of groups or individuals in an area, and/or their activities, or delays in return of people to their affected area; have become more common modi operandi in the logistics of disaster response [37]. These responses typically reflect a manifestation of human values [38] and are intended to improve safety and security, limit further losses, injuries, or deaths; and rebuild or restore impacted infrastructure as quickly as possible with limited interference—or additional damages.

The responses to such an event (e.g., intense rainfall events, [39] is therefore fundamentally different from responses to poor air quality (e.g., an extended smog episode [40]) that may include recognition of a biophysical synergism with the environmental hazard and its ties to various economies. Yet both contain human (or societal) systems interactions that are not well-understood and which impact multiple time domains. They define a complex operational vulnerability of societal responsiveness. What is unclear is how much and how well Smart Systems can assist in widely varying situations, aside from advance preparations and planning of response while integrating these human components. To address these issues, this paper offers an overarching review to explore the

complex set of interactions that exist with regard to the application of Smart Systems, architectures, and applications and tools to the problem of hazards impacts and response.

The interactions of human systems with environmental hazards is first considered (Section 2) with regard to existing and created (or aggravated and compounded) vulnerabilities as related to societal responses. These systems interactions (Section 3) are explored with regard to the complexities involved in managing expectations (of society), standard responses, and the usefulness and true applicability of observational systems in an actionable manner for the decision-making process before, during, and after an environmental hazard. These reveal actionable gaps in existing responses (Section 4) which fundamentally define explicit psycho-social responses to and interactions with environmental hazards and specific emergency responses in time and space. Several components define and contribute to the effectiveness (and failure) of those specific responses both collectively and individually. Thus, the knowledge and integration of these factors and the consequences of actions point to pathways (Section 5) that could be used to tie two disparate systems with and through the use of Societal Smart Systems.

The intent of these pathways is to improve the efficacy of responses before, during, and after environmental hazards while accounting for nonlinear and chaotic psycho-social responses. Several means and approaches of identifying these pathways are explored in an attempt to help direct and orient a substantive transdisciplinary and holistic effort that will make use of the burgeoning body of literature in a longitudinal manner while making use of a variety of experimental methodologies and techniques. These are ultimately intended to guide the conceptualization and design of future interactive and interdependent Smart Systems to ameliorate impacts associated with environmental hazards in a more comprehensive manner that are both adaptable and self-evolving in real-time.

## 2. Vulnerabilities

The vulnerabilities of a population may be empirically or statistically defined (mathematical). They may be evaluated from an emergency response point of view (i.e., physical and societal). However, they measure two very different systems. In the former, demographics (e.g., age, housing, per capita, and other characteristics and empirical data) drive individual and group responses of populations. In the latter the focus is on known quantities (i.e., of hazards) as they pertain to emergency responses and system behaviors associated with infrastructure. In other words, societal response contains varying degrees of uncertainty and may simply be unreliable [41] or unpredictable and chaotic. Public response is a function of engagement, reaction, and response as well as interpretation and acceptance of messages according to perceived risks and experiences. Similar to public response to climate change or general distrust of science [42]; there are perceptions, beliefs, and political systems [43,44]. These create and/or support misunderstanding, misinformation, and misapplication of information [45–47]. In the case of environmental hazards these can increase vulnerabilities.

These societal responses affect initiation of action and must be overcome for both hazard response and in policy development [48,49]. These additional sources of vulnerability, created by the responses of society, are not well measured nor fully incorporated in response mechanisms or in the holistic preparations set in motion when Society faces an environmental hazard. This creates a vulnerability capacity gap [50] due to a lack of understanding and modeling of societal systems with regard to environmental hazards, and according to the various manifestations of a hazard response as related to human behaviors (i.e., advance preparation, reaction, subsequent action, and interaction). Conversely, non-human systems such as infrastructure and their response are inherently and predominantly deterministic with specific and often well-known constraints or thresholds of response. They tend to be driven by the age and capacity of structures, measured against extreme value or skewed distribution approaches [51–53], and climatological information and attribution. They can be "taken off-line" as an at-will protective measure in response to an approaching or anticipated hazard. In some cases their design capacity is exceeded and damage or loss is anticipated in the planning responses.

Yet these deterministic methods (i.e., of the non-human systems such as infrastructure) may not be completely nor adequately designed or appropriate (e.g., defining parameters relative to a Gaussian Distribution) without considering alternative statistical and modeling approaches (e.g., rather than the "100-year flood" model [54–57]). Added to such complexity is the coupled nature of the societal and non-human systems and their interactions (e.g., push and pull dynamics [58]). These are defined by human interventions, responses, and reactions. They imply both constructive and destructive interference among, within, and between systems with many "moving parts" that can produce unexpected or new consequences and effects (positive or negative). Such "shocks" are systemic and make defining a paired interactive system by statistical distributions of response non-normal, nonlinear, and governed by complex multivariate and polynomial relationships. These cannot be readily generalized and applied for all environmental hazards and locations impacted as the initial conditions, statistical constructs, and recurring hazard events are not stationary quantities nor processes. The "shocks" may also generate feedback that is further amplified and which can create new responses and reactions among the systems—these act to further confound examination of the systems, factors, and processes involved.

## 3. Systems Interaction

Therefore, this pairing of Smart Systems with societal and non-human systems in the context of environmental hazards raises the level of latent vulnerability and continually interacts to decrease the efficacy of response. This applies before, during, and after an event and is complicated by the ingest of a non-stable and non-stationary (societal) set of system behaviors (i.e., preparation, reaction, subsequent action, and interaction). Thus the very premise and enterprise of Smart Systems development, deployment, actuation, or application in the context of societal systems and coupled interactions requires an incorporation of psycho-social behaviors of individual and group responses (i.e., preparation, reaction, subsequent action, and interaction). This includes contributions from Environmental Sociology [59] as well as definition of the complexities as driven by Computational Social Science [60]. These behaviors operate on multiple time scales, may be regionalized and localized spatially, and may trend, shift, or swing unexpectedly with the "ingest" of new information and even "distance in time" (i.e., separation time since an event last occurred [61]).

Thus in some respects the problem is analogous to the initialization of an atmospheric model with various perturbations; any one of which may result in a wide variety of solutions based upon deterministic equations [62]. While this presents predictive difficulty (and limitations), the use of ensemble method permutations allows for the creation of a useful forecast product [63] (albeit with a human interface for interpretation and application). The resulting products are intended to serve as guidelines for an effective response. For example, the anticipation of a major hazard that may result in the closure of roadways, cancellation of events, curtailment of aircraft and transit operations, and loss of power informs the planned response [64]. The particular responses and their effectiveness (e.g., real or suggested, such as in validation studies [65]) can then be monitored and reported via news outlets (online or otherwise) before and during impacts. Yet the resulting expectations of Society members may lead to shortages in goods (e.g., water, fuel, and food; [66]) and reduced service capacities (e.g., mass transit, transports; [67]) as well as materials needed to implement preventive measures (e.g., batteries, generators). These collateral societal responses may not be fully included in response planning and delivery.

Expectations are also contextualized according to day of the week, timing during a school-day, and many other permutations of the population and individuals affected [68]. They may reflect limitations in responsiveness based on past experiences of a population (e.g., of forecast errors such as a less intense event or media-hype; [69]). While economically some actions may be productive (i.e., increased sales) such actions can create or amplify multiple recurring costs (e.g., closures of schools and business, un-necessary overtime road crew staffing/pay, emergency personnel activation, move of commercial aircraft to another location, et cetera; [19,70]). During the hazard, which contains varying lifetimes

and sequences of impact (e.g., onset, duration, intensity of a low pressure system, wildfire, dust storm, or tornado) news coverage and online sources can provide multiple observational data, accessible by the general public in raw and processed formats. However, the often disparate data may not be comparable in terms of precision or in that information's applicability to a given location, region, or individual's situation. The usefulness of the disparate data may thus be in doubt when considering the interpretive applications of, or misuse by, individuals and local communities and agencies.

Yet these observational platforms provide data which is clearly considered to be actionable by the public on a highly individualized basis. These are driven by psycho-social interpretations as a function of an individual or group or agent [71] and depends upon their understanding of the data, information, and how it corresponds to a significant threat or danger. This confounding creates greater uncertainty and promotes higher variability in responses. The inclusion of Agent-Based Modeling [72] is thus relevant to examining the processes of, and those associated with, societal response as a collective of individuals. Indeed, an individual's response is driven by conclusions that may not have merit (e.g., assessing own risks) nor a sound-basis given a lack of expertise, data quality and reliability, or reasoning. It may be driven by expectations based upon a non-comprehensive understanding of the specific hazards as a function of the individual's location, mobility, timeliness, and resilience—and what is predicted to occur with the hazard.

This individualized mechanism and process of response colors and reveals societal behaviors as a function of individual self-assessments of vulnerabilities, engagement levels of individuals, resource information available, and forecast information received or accessed. Any of these may be readily aided or exacerbated, but rarely abetted, by Social Media through generation of "echo chambers" (perhaps similar to [73]) or "viral video", "fake news", and by intended or perceived interference (e.g., "Facebook" and elections in the U.S.A.) versus "information overload" [74]. Thus while the interactions of Society in its hazards response as an engaged, pro-active community is beneficial, it also presages unexpected reactions and presupposes—or generally assumes—that appropriate responses will follow. For example, why would someone not evacuate a coastal flood zone given the inherent risks and imminent dangers posed by a powerful cyclone expected to inundate their area? How would that individual truly know the resiliency of the local infrastructures and emergency responses? These questions suggest a paradigm paralysis in planning and response to environmental hazards which in part can explain increased difficulties in the use and promotion of societal responses that are not as effective in protecting life and property as they were in the past.

## 4. Actionable Gaps in Response ("*How i-Verifi*")

To move past such paradigm paralysis requires conceptualization and quantification of the actual vulnerability capacity found within both infrastructure and societal systems. In the context of environmental hazards and their specific components (e.g., physical quantities and events that describe the hazard) it requires an understanding of how the hazards and societal responses change in time and space (i.e., as related to preparation, reaction, subsequent action, and interaction). The coupling of cyber, physical, and human (or societal) systems (i.e., Smart Systems, hazards, and society) with hazard responses is therefore intrinsically nonlinear and non-scalable given variations in hazards and populations. These directly contribute to complex operational vulnerability. Thus growing individual vulnerabilities (i.e., both uncertainty of interactions and the unpredictable behaviors) in human and assisted responses involving Smart Systems may be enhanced by individual engagement, use of resources and resource information available, and forecast interpretation (or viability). These create actionable gaps in response of those at risk and can amplify even minimal risks related to societal engagement responses.

Engagement in the use and understanding of data and resource information that is obtained (rather than received) and interpreted by societal members [75] depends in part upon education and awareness programs. However, engagement also belies the fact that individuals do not have the breadth of experience or tools for decision-making in assessing threats to life and property [76]. The engagement

does not acknowledge inherent fears, interpretation of conflicting or diverging forecasts or statements rather than consensus, (not unlike climate change dialogues [77], or misinformation with regard to "Superintelligence" [78]); the diffusion of information via Social Media [79]; nor the demographic that may mistrust or hold a disbelief in supporting science or governmental offices and their responses (e.g., "sponsored content"). Such reactions are not the same as populations that may be receiving misinformation, misleading statements or analyses, or misapplying the same to their location and timing of their specific actions [80]; and those whom are misdirecting their own efforts to avoid, mitigate, or prevent impacts due to incorrect or faulty assessments of their own risks [81,82].

Variations along demographic lines (e.g., marginalized groups [83,84]) paint a collage of competing interests and vulnerabilities. For example, in the case of a flooded [85,86] or foggy roadway, with or without precautionary information posted during the environmental hazard (plus a news report or correspondent), a pathway to loss of lives and damaged vehicles is directly manifested in societal responses. Indeed, those who survive such encounters often report an inability to effectively gauge or understand the danger (flooding) or to have made a faulty comparison (fog) with their own past experiences of a similar situation [87]. An often heard remark is that an individual "never saw anything like" or "never before experienced" the hazard encountered [88,89]. Similar principles apply to an individual not evacuating in advance of a powerful tropical cyclone because they have "lived through" prior events. They may simply be overestimating their resilience in withstanding multiple and varied components, combinations, and impacts of a cyclone threat—or be skeptical of the forecast information, and mixed messages, provided through various media [90].

In the same sense, deployment of a Smart System alert for an underpass or causeway that floods may in fact be insufficient in deterring a potential victim from venturing into a dangerous situation. Alternatively, an individual may decline evacuation; yet in the face of a major environmental hazard later request extraction after monitoring Smart System data. Unfortunately, that request may occur at a time when emergency response is no longer feasible due to the dangerous conditions and thus an individual's vulnerability capacity gap is realized. These examples reflect individual and societal response and awareness spectra characterized as nonlinear—whether those responses are appropriate to the hazard or not. Thus while "best practices" may be applied in terms of environmental hazards planning and delivery, identification of overall effectiveness relies upon the underlying engagement level of the populace and their collective and individual reasoning and judgement—and these are non-stationary in time, space, and by phenomenon or hazard.

The omnipresence of resource data and information through Smart Systems, and the use of forecast interpretation (and its viability) provide additional, interdependent, and confounding multivariate functions. Collectively they describe and define individual and societal responses to hazards. In the case of Smart System data and AI, the availability and timeliness of these are potentially critical to emergency planning and have been documented with regard to in situ response during an event (e.g., an elderly population threatened by fire or flood in a Smart Home or facility [91]). Indeed, people-centric responsive architectures have already been proposed (e.g., [92]) but may lack comprehensive and overarching coverage when applied to a collection of diverse communities that necessarily interact with one another. Yet the same will still be unreliable for a population of users unable to account for timing differences and the limitations of data sampling relative to their region (e.g., watershed, flooding). The same users cannot properly assess the threshold values that depict the level of danger that is not predicated upon individual response and interpretation or interpretive capability. In such situations an alert by local governments [93] and responders may be "ignored" or interpreted as to "not apply" by an individual.

Similarly situations may be encountered in advance of a hazard by way of forecast information that has inherent errors in terms of location, timing, and intensity of an anticipated hazard. When such predictive limitations are combined with a user's inability to properly assess the dangers, societal susceptibility increases in terms of applicability of forecast information due to coverage (and location) issues. *Individual vulnerability* then can be defined by a person's *engagement* with specific *resource*

*information* and *forecast interpretation*—or on a more personal level, "*How i-verifi*" an environmental hazard or threat. In the flooding case above, inclusion of duration and hydrologic flow parameters that are either unknown to an individual, misapplied based upon past experiences, or considered to not apply to them explicitly—and varying confidence levels in those forecast parameters—are problematic. Additionally, the saliency of data (or imagery) meant to elicit response may be modeled according to an individual's visual attention [94] or acuity (i.e., ability to render relevance and reaction) to predict or mimic the actual responses expected. Thus whereas forecast confidence levels and uncertainty present a danger given their intent versus their interpretation and understanding by the public they can result in errant decision-making given dependence upon the event types encountered [95].

As an individual's ability to ascertain danger, risk, and make an appropriate decision may be significantly flawed, a Smart System's emulation of that same capability (or lack thereof) is required. This requires an adaptive understanding of psycho-social behaviors relative to and intertwined with physical systems based upon hazard response recommendations of planners, emergency managers, and government officials. These define the actionable gaps in response by individuals (or "*How i-verifi*") that complicate and reduce efficacies in planning, real-time use of Smart Systems, and overall effectiveness of responses intended to protect life and limit damage to property (and infrastructure). They speak to complexity in terms of hazard response coverage (over time and in space), engaging individuals in responses that could allow for evaluation of the efficacy in identifying and resolving an individual's actionable gaps in response. This complexity increases for populations presented with multiple scenarios (or "what if" situations). Whether a message for an approaching hazard or changes in hazards as related to climate change [96–99], the inclusion of anticipated, real-time, and variable psycho-social behaviors is extremely challenging.

## 5. Shaping Pathways (Moving from "How i-Verifi" to COVERAGE by Societal Smart Systems)

The psycho-social reaction of individuals and an entire population to environmental hazards have received much attention and have been contextualized in real-time and post-facto [100]. However, maximizing model manifestations and incorporating such random and chaotic elements has deterministic limits in predictability. These are clearly relevant to planning, preparation, endurance, and aftermath behaviors associated with environmental hazards. They highlight the significance of identifying the issues from an individual basis as above (i.e., *How i-verifi*) and the relevance of interactions among the aggregate communities affected and their use of Smart Systems. The complex set of interactions and the confounding response solutions that arise across individuals and according to regional variations in such coverage; in spite of best practices, forecast data availability, and the use and application of real-time in situ observations by Smart Systems are daunting if not overwhelming problems for emergency response. In tandem these context specific aspects address the need to re-examine decision making; and the auto-reactive and auto-correlative individual and community responses associated with environmental hazards.

To a certain extent the level or measure of perceived peril by an individual may be considered using psycho-social cues in Social Media. These can allow comparison among groups only when normalized according to demographic aggregates to be able to differentiate true responses from those that may present as humor and sarcasm or other effusions (e.g., [101]). Clearly these lack scalability and comparability and are nonlinear systems and responses that are non-deterministic, or more simply are best represented as probabilistic quantities. In other words, defining *Complex Operational Vulnerability* requires *Engagement Responses* (interaction terms) that identify *Actionable Gaps* (and auto-reactive components) in those responses to improve *Efficacy* (*COVERAGE*) in the application of Societal Smart Systems (and thus overall response). This could be accomplished by maximization in modeling of societal responses to environmental hazards with a manifestation of Societal Smart Systems information, data, and predictions. This would improve "*COVERAGE*" of societal systems and their human components and responses before, during, and after hazards by creating pathways to reduce vulnerabilities.

To create and "shape" the pathways of maximum *COVERAGE* a significant set of parametric and non-parametric correlative relationships could be derived, not only from the fundamental principles of emergency response, sustainability, and vulnerability; but with the partitioning and inclusion of confounding factors, positive and negative feedbacks, and nonlinear responses expressed probabilistically. The approach allows for coverage of widely varying hazards and responses; impacts the need to account for multimodal actions and reactions, and considers specific initialization and assimilation modes associated with that used in Smart Systems architectures. It allows for cross-scale application and consideration of localized as well as more generalized effects and impacts. This is in accordance with the Internet of Things Paradigm whether an urban [102] or other setting with attendant simulations and tools—and could be applied to issues surrounding environmental hazards.

With deterministic and forecast information, specific scenarios could be created or explored with explicit inclusion of the limits of predictability of interactive systems. However, rather than planning only scenarios, or asking "what if" questions, it could allow for specific response strategies to be found—not simply formulated or enumerated—with well-defined confidence limits. Such response strategies are more compelling as they could be applied ad hoc to create a set of self-evolving or even self-generating solutions and displayed as an array of choices by an end-user in real-time [103]. Allowance for and inclusion of psycho-social responses of populations affected by specific hazards—according to location, timing, and demographic expectations—could then be completed without recourse to a predetermined pathway, or predetermined best practices applied routinely or out of context. This approach shares similarities with latent semantic analysis [104,105], event extraction [106], and other techniques (e.g., use of semantic observations [107]) and therefore presents a pathway to understanding and modeling both auto-reactive behaviors and specific emergency response actions before, during, and after an event.

It allows for application among varying regional population demographics and that of specific individuals. The use of Social Media could even incorporate "emoji" and "meme" constructs that provide direct insight, just as time and location stamps do, as the expressions relate the current and evolving psycho-social state according to an individual's context and circumstance in the midst of an environmental hazards' observed impacts and physical manifestations. This approach allows for a reshaping of the discourse (i.e., not unlike moving beyond the "environment versus economy" debate) by understanding the values, concerns, and positions of interest by a population's members [108]. It can thus produce a "best-response" by individuals to the declining resilience and increased vulnerability and risk associated with environmental hazards and specific events. The delivery of societal and individual responses in a matrix of probabilities (before, during, and after an event) could also be applied across specific urban and suburban locations to better depict types and levels of response, impacts, actions, and reactions. The depiction could include expression of the limits to predictability and thus the confidence levels of various responses (and their efficacy).

Just as with attendant use of communications resources during a hazard by individuals who are contextually aware; the same context awareness can aid and improve Smart City applications and resource allocation decision-making [109]. These could be shaped according to a hazard and its inherent variations (e.g., to include more frequently occurring non-tropical cyclones [110]) in terms of repeatability and its numerical and operational modeling. Thus the use of localized information (e.g., landscape, infrastructure, land use, food supply, and transportation or fuel systems) according to deterministic values (i.e., quantity, quality, robustness, critical values, and more) that change in time and space and in response to societal interactions could be layered into urban and suburban areas to depict changes as a function of Smart System data and information. This is similar to constructs examining city heat amplification by factor separation [111] and the FACETS Paradigm approach in weather forecasting that employs probabilistic and forecast information [112,113] for direct application by various user communities.

The approaches envisioned here need not rely upon a limited statistical construct (e.g., "100-year flood") but rather could be used to better inform and direct [114] societal responses to become the

most practical, pertinent, and appropriate during an encounter with an environmental hazard over a span of time (before, during, and after). These could also be applied across a hazard's multiple array of impacts. Such "moving targets" can only be managed effectively with inclusion of human and non-human systems data, information, and responses as incorporated in the real-time and organic process of response. A set of response solutions can more reliably identify likely and unexpected or chaotic reactions (e.g., a nodal point) and portray the role of self-organized criticality (e.g., citizens helping one another). Alternatively orbital pathways or nodal points of response (i.e., deterministic chaos in mathematical and physical sciences) can be determined by incorporation of the psycho-social aspects already defined in the burgeoning body of literature across multiple disciplines. Achieving *COVERAGE* is essential to and informs hazard impact mitigation and prevention strategies that support resilience.

The foregoing includes responses that may be rare or reach an end-point (final state) that leaves no further viable recourse or solution such as a "dead-end" (i.e., destruction of life and property) based upon the hazard, response, and individual. Identification of "hot-spot" issues, developing situations across a region, and changes in emergency responses—while in motion—suggest further activation of data mining in generation of Smart System data applications and use of Social Media [115] to avoid unpleasant outcomes. Indeed, the use of drones away from urban zones [116] in such cases could provide direct in situ observation and analysis of infrastructures (e.g., power facilities, rail and bridge systems) and supply lines—or escape routes. Use of calculated and deterministic factors and taking advantage of probabilistic methods would provide and promote guidelines development to better shape the pathways of Smart Systems in terms of structure, architecture, usage, and reliability with societal behaviors inherently present.

Calculated and deterministic factors may also provide greater insight and invite public participation in policy design and implementation as a function of computational social science applied to environmental hazards. This could include use of a general theory of scientific/intellectual movements (SIMs; [117]) in guiding the discourse. Planning by a Smart Society taking advantage of services geared towards Smart Cities [118] and situations of personal interest (e.g., as reflected by Social Media) could assist end users in their manipulation of data, information, and Social Network information resources [119]. Indeed, Smart Systems with the capacity to factor-in local inhabitants' metadata and their specific characteristics (e.g., owners of chainsaws, volunteer fire-fighters, and other items or skill-sets) would quickly highlight points of contact in a community relevant to recovery efforts (with ethical and privacy concerns properly considered). These are key facets that are, and continue to be reflective of, populations and individuals that reach-out to help and support those who are affected by environmental hazards—whether before, during, or after—and in the aftermath of impacts that are individualized yet community-oriented.

The pathways presented here are intended to guide the conceptualization, design, and delivery of new interactive and interdependent systems to provide even greater benefit from Smart Systems—particularly with regard to environmental hazards and societal responses.

**Author Contributions:** The conceptualization, review of the literature, and analysis of this investigation were conducted directly by the author.

**Acknowledgments:** Efforts of investigators spanning many disciplines made this paper possible.

**Conflicts of Interest:** The authors declare no conflict of interest.

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
