# Peer review of "Environmental Hazards: A Coverage Response Approach"

_futureinternet, doi:10.3390/fi11030072_

Round 1

Reviewer 1 Report

Overall, this is a timely thought piece. The paper explores the issue of smart systems and how they can be used prior, during and after hazardous events. The literature is comprehensive and is not just focused on the systems of system approach but also the people aspect of smart systems which is essential in my opinion if we are to advance well being.

While there is a lot of work on smart systems, often the people aspect is overlooked or how people might utilize or ignore such systems. This is not the case with this paper. And therefore I feel this is a valuable contribution.

That being said, I do I have questions and suggestions that I feel would improve the quality of the paper.

In the abstract the paper writes “the use and application of real-time in situ observational platforms” and … “relevant architectures, and ever-increasing numbers of applications and tools development by individual” What would be good would be to give examples of such platforms, architectures and applications from both government agencies and say form the people/crowdsourcing (i.e. VGI etc). this would help later in the “How i-verifi” and “COVERAGE” sections. In the sense, it would show why these sections are needed and what is currently missing form in situ observational platforms, architectures, and platforms. Morover, terms used in later sections especially with respect to COVERAGE need to be introduced in ealier sections.

For example, there are examples of how the USGS is using twitter for earthquakes (e.g. Earle, P. S., Bowden, D. C., & Guy, M. (2012). Twitter earthquake detection: earthquake monitoring in a social world. Annals of Geophysics, 54(6).) Similarly twitter, flickr have been used for floods and wildfires.

e.g. Liu, S.B.; Palen, L.; Sutton, J.; Hughes, A.L.; Vieweg, S. In search of the bigger picture: The emergent role of on-line photo sharing in times of disaster. In Proceedings of the Information Systems for Crisis Response and Management Conference, Washington, DC, 2008.

Vieweg, S.; Hughes, A.L.; Starbird, K.; Palen, L. Microblogging during two natural hazards events: What twitter may contribute to situational awareness. In Proceedings of the 28th International Conference on Human Factors in Computing Systems, Atlanta, GE, 2010; pp 1079-1088.

This issue is seen thrughtout the paper. For example in Section 3 you write “The particular responses and their effectiveness (real or suggested) can then be monitored and reported via news outlets (online or otherwise) before and during impacts.” There are examples where this is happening and therefore it could be supported with examples applications and references.  Same goes for line 125 to 136, your argument is devoid of references to support your discussion.

Also your choice of reference 50 is strange in the sense you are referring to an agent-based model that does not use any real world data but only explores politics abstractly. What is needed is a better rationale for why you are citing agent-based modeling (ABM) and how has agent-based modeling has been used to explore environmental hazards (e.g. earthquakes, floods, wild fires etc) which there are numerous examples. One could argue that ABM offers a different way to explore these from the bottom up but in the current paper this is missing. If the purpose is focus on people and coupling as you allude to in Section 4 your argument could be more tightly interwoven.

Lines 74 to 78 could have some references to support the discussion.

In Section 4 you talk about “How i-verifi” which is a nice idea but the concept could be more clearly defined. For example, after reading this section numerous times I am still wondering what a example would this look like and how it could be implemented and this therefore makes its link to COVERAGE hard to grasp. Also the words that make up COVERAGE:  “Complex Operational Vulnerability requires Engagement Responses (interaction terms) that identify Actionable Gaps (and auto-reactive components)” do not really come out in the previous sections. What would be good is that in the sections leading to this that the terms in the acronym where specifically discussed

Also in Section 5 as you are proposing COVERAGE should the tense not be “could be” rather than “can be” or “it allows”. I.e. Tone down what COVERAGE could be used for?

A minor issue you write “and the FACETS Paradigm approach employing probabilistic and forecast information” but you don’t provide the reader any idea on what its for. For example, weather forecasting. I.e. please don’t leave the reader hanging.

Minor issues:

Line 28 the pew report should be referenced like the others.

Line 47 the term “volunteered geographic information” should also have the reference to Goodchild who coined the term. Ie. Goodchild, M. F. (2007). Citizens as sensors: the world of volunteered geography. GeoJournal, 69(4), 211-221.

Line 315  you use the term “Social Web” for the first time. Should it not be another word or should the concept not be introduced earlier?

Reviewer 2 Report

This is a essay of considerable merit discussing a situation that relates well to the issues confronting response agencies, and is supported by a comprehensive review of the literature. 

However and as a reviewer I was confronted by two issues:

The editorial style is complex with many issues often addressed in the one sentence. This may be a personal preference for editorial style which I accept and present only as a suggestion to enhance the engagement with the audience for this research. I have included my notes for the first sections of the essay for you information which demonstrate the process I adopted to ascertain the focus of certain parts of the essay

The report falls short of providing possible applications of the resource created by the literature search. In my opinion, the research project is only half complete. While the description of the situation is complete, the hypothesis/hypotheses are not tested and nor are there findings that are sufficiently definitive to guide those researching effective response measures

It may be that the editors are satisfied with the publication of an essay discussing the situation and if that was the case the essay would be of interest to some, but in my opinion the research does not progress far enough to advance to the body of academic knowledge on the subject

Round 2

Reviewer 1 Report

Many thanks for addressing my comments from the first round of reviews, which I believe helps better position this paper. Upon rereading the paper, I have nothing major to say. However, if the author was inclined I would suggest in the introduction being explicit about the purpose of the paper and giving the readers a brief outline including section numbers (i.e. so they know what to expect and how this relates to the purpose of the paper).

Reviewer 2 Report

Thanks for considering my comments

Round 3

Reviewer 2 Report

Thank you for the consideration of my comments that go some way to addressing the issues raised. 

Accepting this essay is intended is a discussion about the wide range of issues that impact disaster response management, and the positive and negative influences of smart systems, it remains an extremely difficult text to comprehend. I also accept that others may not be as troubled by the language but, that said the fundamental problem remains the jumping across concepts, the drawing of conclusions unrelated to the discussion, the interchangability of smart systems, social media etc, even the linking of specific disaster responses and responses to climate change. The last 9 paragraphs are a very good discussion of the issue, the preparatory discussion of the confounding issues could equally be be as pithy, removing the extraneous and often contradictory explanations. For example, at line 405 you conclude ", Smart Systems (have) the capacity to factor-in local inhabitants’ metadata  ...", whereas at line 222 you  contradict this by saying  "The  coupling  of  cyber,  physical, and  human  (or  societal)  systems  is  intrinsically   non-linear   and   non-scalable   and   directly   contribute   to   complex   operational vulnerability", a point you make many times.  

Attached is my comments to specific issues of language and the logic of some statements.
